# `DGExplainer`: Explaining Dynamic Graph Neural Networks via Relevance Back-propagation

## Abstract

Dynamic graph neural networks (dynamic GNNs) have demonstrated remarkable effectiveness in analyzing time-varying graph-structured data. However, their black-box nature often hinders users from understanding their predictions, which can limit their applications. In recent years, there has been a surge in research aimed at explaining GNNs, but most studies have focused on static graphs, leaving the explanation of dynamic GNNs relatively unexplored. Explaining dynamic GNNs presents a unique challenge due to their complex spatial and temporal structures. As a result, existing approaches designed for explaining static graphs are not directly applicable to dynamic graphs because they ignore temporal dependencies among graphs. To address this issue, we propose `DGExplainer`, which offers a reliable explanation of dynamic GNN predictions. `DGExplainer` utilizes the relevance back-propagation technique both time-wise and layer-wise. Specifically, it incorporates temporal information by computing the relevance of node representations along the inverse of the time evolution. Additionally, for each time step, it calculates layer-wise relevance from a graph-based module by redistributing the relevance of node representations along the back-propagation path. Quantitative and qualitative experimental results on six real-world datasets demonstrate the effectiveness of `DGExplainer` in identifying important nodes for link prediction and node regression in dynamic GNNs.

## 1 Introduction

Dynamic GNNs have achieved significant success in practical applications such as social network analysis (Zhu et al., 2016), transportation forecasting (Bui et al., 2022), pandemic forecasting (Kapoor et al., 2020), and recommender systems (Zhang et al., 2022). However, since most of the dynamic GNNs (Ma et al., 2020; Li et al., 2017; Nguyen et al.; Goyal et al., 2018; Yu et al., 2018a; Seo et al., 2018; Hajiramezanali et al., 2019) are developed without interpretability, they are treated as black-boxes. Without understanding the underlying mechanisms behind their predictions, dynamic GNNs cannot be fully trusted, preventing their use in critical applications. In order to safely and trustfully employ dynamic GNN models, it is important to provide both accurate predictions and human-understandable explanations.

The explanation techniques for static GNNs have been extensively explored by recent studies. These techniques include approximation-based methods (Baldassarre & Azizpour, 2019; Pope et al., 2019), which use gradients or surrogate functions to approximate the output of a local model. Perturbation-based approaches (Ying et al., 2019; Luo et al., 2020) explain static GNNs by masking specific features to observe their impact on the model's output. Gradient-based methods (Sundararajan et al., 2017; Selvaraju et al., 2017) adopt the additive assumption of feature values or gradients to measure the importance of input features. Further relevant research on explaining static GNNs can be found in Section 4.5.2. However, these methods do not account for the unique temporal information essential for explaining dynamic GNNs. Directly applying existing explanation frameworks for static graphs to dynamic graphs is impractical, as it leads to discrete explanations for a graph sequence, with each graph snapshot being explained independently.

Explaining dynamic GNNs can be challenging. We illustrate this process in Figure 1. The prediction task, shown in Figure 1a, aims to forecast future traffic flows (denoted by dashed lines) at different locations based

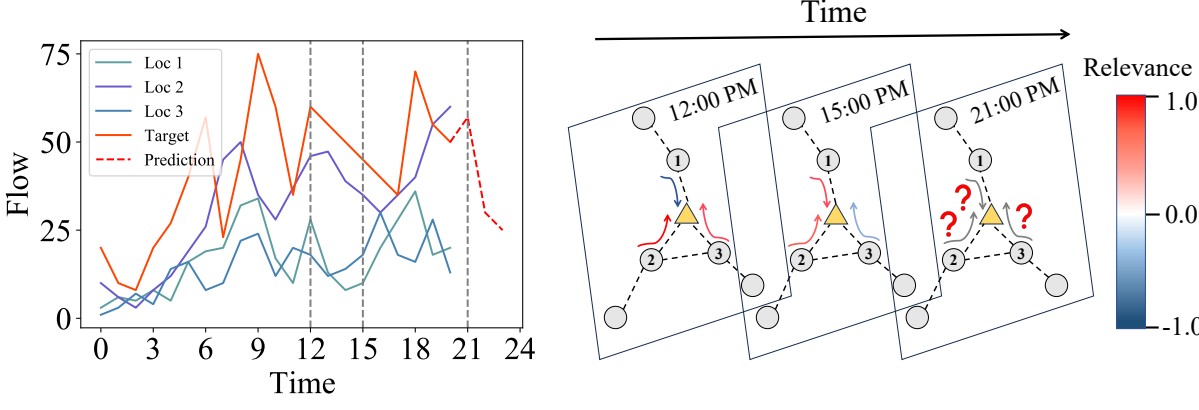

(a) Traffic flow prediction.     (b) Explaining traffic-flow prediction of dynamic GNNs.

Figure 1: The diagram of the explanation task of dynamic GNNs on traffic flow data.

on historical observations (denoted by solid lines). This spatial-temporal data is modeled as a dynamic graph, represented in Figure 1b, where each graph snapshot records traffic flows at different time steps (e.g., 12:00 PM, 3:00 PM, and 9:00 PM). In each graph snapshot, a dashed line between two nodes indicates a commute between locations, and an arrow represents traffic flows, contributing to the prediction for the target location (denoted by a yellow triangle). The explanation task aims to determine the influence of other locations on the prediction of the target location. The polarity of the influence is denoted by the color of the arrows: blue indicates a positive correlation, while red indicates a negative correlation, with the darkness of the color indicating the strength of the influence. The complexity of dynamic graph data necessitates both temporal and spatial module designs in dynamic GNNs. This makes the explanation task challenging, as it requires identifying the influence of the input based on the output from these dynamic GNNs.

To integrate unique temporal and spatial information in explaining dynamic GNNs, we propose using layer-wise relevance propagation (LRP). Originally introduced by Bach et al. (2015) for image classifiers, LRP computes the relevance of each pixel in predicting an instance. Applying LRP to dynamic GNNs offers two key benefits. First, unlike most explanation techniques for static GNNs, it does not require learning a surrogate function or running any optimization procedure. Second, LRP evaluates the importance of sequences of edges or walks in the graph, rather than focusing solely on individual nodes or edges, making it particularly well-suited for explaining dynamic GNNs.

To address this challenge, we propose a framework called `DGExplainer` (Dynamic Graph Neural Network Explainer). The framework operates in three main steps. First, it decomposes the prediction of a dynamic GNN and computes the relevance in a time-related module using relevance back-propagation. Second, it calculates the relevance of the input features by back-propagating through the graph-related modules (e.g., a GCN module) layer by layer at each time step. Finally, by aggregating the relevance from the previous steps, we obtain the final relevance of node features, which represents their importance to the prediction. The contributions of our work are as follows:

- This work aims to explain the predictions of dynamic graph neural networks, marking one of the pioneering efforts to tackle this challenge.

- We propose a novel framework, `DGExplainer`, designed to generate explanations for dynamic GNNs from a decomposition perspective. `DGExplainer` effectively calculates relevances that represent the contributions of each component in a dynamic graph.

- We demonstrate the effectiveness of `DGExplainer` on six real-world datasets. Quantitative experiments across three evaluation metrics show that our method provides faithful explanations. Furthermore, qualitative experiments demonstrate that `DGExplainer` offers significant advantages over other baseline methods in effectively explaining dynamic GNNs.

## 2  Problem Definition

This paper focuses on explaining dynamic GNNs by computing the relevances of input features. Our approach first redistributes the prediction from the last layer to the relevances of hidden representations. Then, we use Layer-wise Relevance Propagation (LRP) to back-propagate the relevances through *time-related* and *graph-related* modules, finally reaching the input layer to obtain the relevances of input features of each node. Our method considers both the structural and temporal information of the dynamic graphs. The relevant notations will be introduced in the following sections.

We study a series of input graphs $\mathcal{G} = \{\mathbf{X}_t, \mathbf{A}_t\}_{t=1}^T$, where $T$ is the length of the sequence. Each graph at time $t$, $\mathcal{G}_t = \{\mathbf{X}_t, \mathbf{A}_t\}$, consists of a feature matrix $\mathbf{X}_t \in \mathbb{R}^{N \times D}$ and an adjacency matrix $\mathbf{A}_t \in \mathbb{R}^{N \times N}$. Here, $N = |\mathcal{V}_t|$ denotes the number of nodes, and $D$ is the feature dimension. The feature vector for node $i$ at time $t$ is $\mathbf{x}_t^i = (\mathbf{X}_t^{(i,:)})^\top \in \mathbb{R}^D$, which corresponds to the $i$-th row of $\mathbf{X}_t$. Without loss of generality, $\mathbf{A}^{(i,j)}$ denotes the entry at the $i$-th row and $j$-th column of the adjacency matrix $\mathbf{A}$, and $\mathbf{x}^{(i)}$ denotes the $i$-th entry of the vector $\mathbf{x}$. The relevance of an element $k$, which can be a node, an edge, a feature, etc., is represented by $R_k$. Additionally, $R_{k_1 \leftarrow k_2}$ denotes the relevance of $k_1$ distributed from $k_2$. The *problem* of explaining dynamic GNNs involves identifying the subgraph within $\mathcal{G}$, which consists of nodes and edges that are most important at a specific time step $t$, given a dynamic GNN model $f(\mathcal{G})$.

## 3  Explaining dynamic GNNs via `DGExplainer`

This section introduces our proposed method, `DGExplainer`, which explains the prediction of dynamic GNNs by back-propagating relevances through both *time-varying* and *message-passing* reverse paths. `DGExplainer` computes the relevance of each input feature by considering both the structural and temporal information of the dynamic graphs.

### 3.1  Preliminaries

In the following subsections, we present the preliminaries relevant to our proposed method. We begin with an overview of Dynamic Graph Neural Networks in Section 3.1.1, followed by an introduction to Layer-wise Relevance Propagation in Section 3.1.2.

#### 3.1.1  Dynamic Graph Neural Networks

Dynamic GNNs (Skarding et al., 2021; Zhang et al., 2022) take a sequence of graphs as input and output representations of topology, nodes, and/or edges. A notable approach involves co-training a GNN with a recurrent neural network (RNN), referred to as a GNN-RNN model. Examples include GCN-GRU (Zhao et al., 2019), ChebNet-LSTM (Seo et al., 2018), and GCN-RNN (Pareja et al., 2020). Detailed related work about dynamic GNNs can be found in Section 4.5.1. Despite the introduction of various methods, recent approaches still do not consistently outperform the GCN-GRU model (Pareja et al., 2020). Therefore, in this work, we choose to use the GCN-GRU model as the basis for elaborating our method. In addition to explaining the GCN-GRU model, we also apply `DGExplainer` to other dynamic GNNs that utilize different GNN or RNN architectures. Detailed results of these experiments can be found in Appendix A.5.

**The GCN-GRU Model–Forward Pass:** In the GCN-GRU model, the GCN module first encodes input features of the current time step, capturing dependencies between nodes. These encoded features are then passed to the GRU module, which captures temporal dependencies across different time steps. Below, we outline the forward process of the GCN-GRU.

**(a) The Graph Convolutional Network (GCN) module:** GCNs represent a node using local information from its surrounding neighbors (Kipf & Welling, 2016). This graph convolution process is formulated as follows:

$$\mathbf{F}_t^{(l+1)} = \sigma(\mathbf{V}_t \mathbf{F}_t^{(l)} \mathbf{W}_t^{(l)}). \tag{1}$$

Here, $\mathbf{V}_t := \tilde{\mathbf{D}}_t^{-\frac{1}{2}} \tilde{\mathbf{A}}_t \tilde{\mathbf{D}}_t^{-\frac{1}{2}}$ is the normalized adjacency matrix, where $\tilde{\mathbf{A}}_t = \mathbf{A}_t + \mathbf{I}_N$ and $\tilde{\mathbf{D}}_t = \mathbf{D}_t + \mathbf{I}_N$. The matrix $\mathbf{D}_t$ is the degree matrix, defined as $\mathbf{D}_t^{(i,i)} = \sum_j \mathbf{A}_t^{(i,j)}$, and $\mathbf{I}_N$ is an identity matrix of size $N$. The

output at the $l$-th layer is denoted as $\mathbf{F}_t^{(l)}$, with the initial layer output $\mathbf{F}_t^{(0)} = \mathbf{X}_t$. Assuming the GCN has $L$ layers, the final node representation at time step $t$, which contains the graph structural information, is denoted as $\hat{\mathbf{X}}_t = \mathbf{F}_t^{(L)}$. The GCN-encoded features from all time steps $\{\hat{\mathbf{X}}_t\}_{t=1}^T$ are then fed into a GRU.

**(b) The Gated Recurrent Unit (GRU) module:** GRU is a variant of the RNN designed to learn long-term dependencies using two selective gates (Cho et al., 2014). In the GRU, each cell processes an input $\hat{\mathbf{x}}_t = (\hat{\mathbf{X}}_t^{(i,:)})^\top$ and a hidden state $\mathbf{h}_t = (\mathbf{H}_t^{(i,:)})^\top$. The update rule for a GRU cell is as follows:

$$\mathbf{r} = \sigma\left(\mathbf{W}_{ir}\hat{\mathbf{x}}_t + \mathbf{b}_{ir} + \mathbf{W}_{hr}\mathbf{h}_{t-1} + \mathbf{b}_{hr}\right), \tag{2a}$$

$$\mathbf{z} = \sigma\left(\mathbf{W}_{iz}\hat{\mathbf{x}}_t + \mathbf{b}_{iz} + \mathbf{W}_{hz}\mathbf{h}_{t-1} + \mathbf{b}_{hz}\right), \tag{2b}$$

$$\mathbf{n} = \tanh\left(\mathbf{W}_{in}\hat{\mathbf{x}}_t + \mathbf{b}_{in} + \mathbf{r} \odot \left(\mathbf{W}_{hn}\mathbf{h}_{t-1} + \mathbf{b}_{hn}\right)\right), \tag{2c}$$

$$\mathbf{h}_t = (1 - \mathbf{z}) \odot \mathbf{h}_{t-1} + \mathbf{z} \odot \mathbf{n}, \tag{2d}$$

where $\mathbf{W}_{ir}, \mathbf{W}_{hr}, \mathbf{W}_{hz}, \mathbf{W}_{in}, \mathbf{W}_{hn}, \mathbf{b}_{ir}, \mathbf{b}_{hr}, \mathbf{b}_{hz}, \mathbf{b}_{in}, \mathbf{b}_{hn}$ are learnable parameters in GRU, $\sigma(\cdot)$ denotes the activation function, and $\odot$ stands for an element-wise product operation.

### 3.1.2 Layer-wise Relevance Propagation

Layer-wise Relevance Propagation (LRP) was first proposed to explain image classifiers by inferring the pixel-wise relevance of an input image (Bach et al., 2015). This method can be extended to other neural networks, such as GNNs. In the following sections, we introduce the original concept of LRP.

Given an image $x$ and a classifier $f(\cdot)$ the aim of layer-wise relevance propagation is to assign each pixel $p$ of $x$ a pixel-wise relevance $R_p^{(1)}$ such that:

$$f(x) \approx \sum_p R_p^{(1)}. \tag{3}$$

Pixels $p$ with $R_p^{(1)} < 0$ contain evidence against the presence of a class, while $R_p^{(1)} > 0$ is considered as evidence for the presence of a class. These pixel-wise relevances can be visualized as an image called a heatmap. Obviously, many possible decompositions exist that satisfy Equation (3). One work yields pixel-wise decompositions that are consistent with evaluation measures and human intuition.

The objective described in Equation (3) can be easily extended to tasks beyond image classification. For instance, in this paper, we study the node classification task where $f(\cdot)$ represents a GNN or dynamic GNN. Here, the goal is to compute the relevance of each feature $p$ for every node $x$ as specified in Equation (3). Further details on applying Equation (3) to other tasks are provided in Appendix A.4.

### 3.2 The Proposed `DGExplainer` for Explaining Dynamic Graphs

With the aim of identifying the important subset of node features that contribute to the prediction of dynamic GNNs, we propose the `DGExplainer`. This method explains dynamic GNNs using the Layer-wise Relevance Propagation (LRP) technique introduced in Section 3.1.2. Similar to many backward-based explanation methods (Schnake et al., 2021; Bach et al., 2015; Pope et al., 2019), it calculates relevances within the range of $(-1, 1)$ to determine the extent to which each component of the model influences the prediction.

In this paper, we consider neural networks consisting of layers of neurons. The output $x_{k_2}$ of a neuron $k_2$ is a non-linear activation function $g$ as given by:

$$x_{k_2} = g\left(\sum_{k_1} w_{k_1 k_2} x_{k_1} + b\right) \tag{4}$$

Assume that we know the relevance $R_{k_2}^{(l+1)}$ of a neuron $k_2$ at network layer $(l+1)$ for the classification decision $f(x)$, then we like to decompose this relevance into messages $R_{k_1 \leftarrow k_2}^{(l,l+1)}$ sent to those neurons $k_1$ at

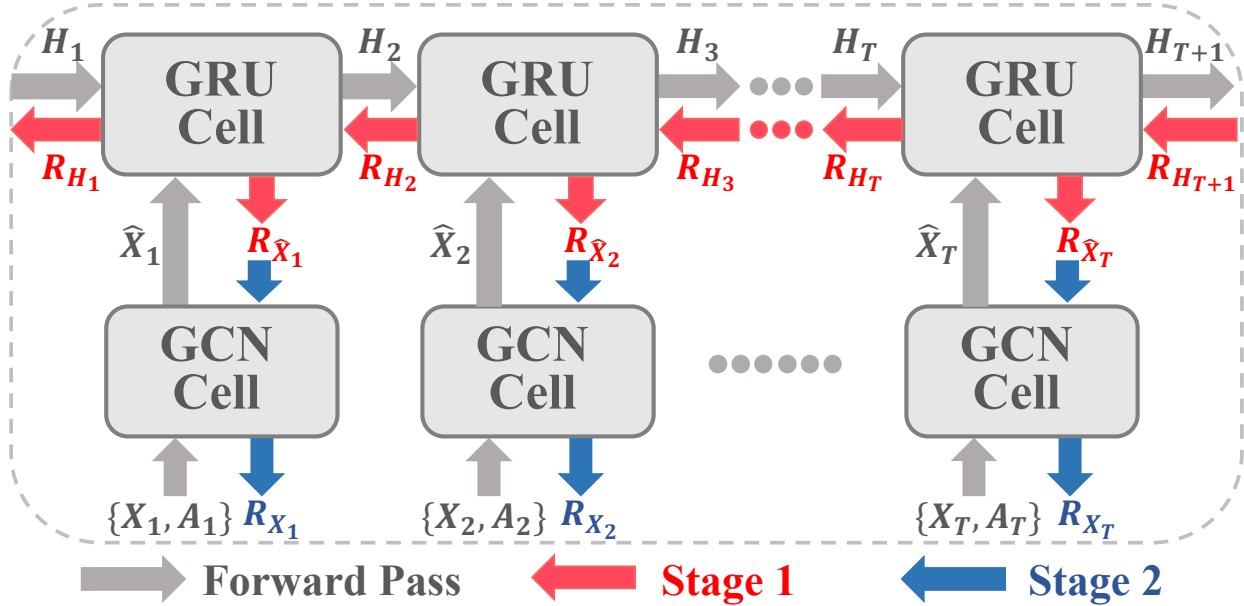

Figure 2: The network structure of the GCN-GRU model and the back-propagation of the relevances. Note that the GRU cells and GCN cells share the same parameters. $\{\mathbf{H}_t\}_{t=1}^{T+1}$, $\{\mathbf{X}_t\}_{t=1}^{T}$, $\{\hat{\mathbf{X}}_t\}_{t=1}^{T}$, $\{\mathbf{A}_t\}_{t=1}^{T}$ represent the hidden states, input features, GCN-encoded features, and adjacency matrices at different time steps, respectively.

the layer $l$ which provide inputs to neuron $k_2$ such that Equation (5) holds.

$$R_{k_2}^{(l+1)} = \sum_{k_1 \in (l)} R_{k_1 \leftarrow k_2}^{(l,l+1)}. \tag{5}$$

We can then define the relevance of a neuron $k_1$ at layer $l$ by summing all messages from neurons at layer $(l+1)$ as in Equation (6):

$$R_{k_1}^{(l)} = \sum_{k_2 \in (l+1)} R_{k_1 \leftarrow k_2}^{(l,l+1)}, \tag{6}$$

The propagation of relevance from layer $(l+1)$ to layer $l$ is defined in Equation (5) and Equation (6). The relevance of the output neuron at layer $M$ is $R_1^{(M)} = f(x)$. The pixel-wise scores are the resulting relevances of the input neurons $R_d^{(1)}$.

To AE QYEs: Q1

`DGExplainer` includes two stages. **Stage 1:** *Compute the Relevances in GRU*. In this stage, the relevances are back-propagated in the time-related module (GRU) along the time-varying reverse paths. This stage aims at computing the relevances of the GCN-encoded features for all time steps. **Stage 2:** *Back-Propagate the Relevances in GCN*. In this stage, the relevances obtained from Stage 1 are taken as input. These relevances are then back-propagated in the graph-related module along the message-passing inverse path. The objective of this stage is to determine the relevance of the input features across all time steps, providing an explanation for the input dynamic graphs. We elaborate on the `DGExplainer` framework in Figure 2.

### 3.2.1 Stage 1: Compute the Relevances in GRU

**Stage 1** focuses on deriving the relevances of the inputs from the relevance of the output for each GRU cell along the inverse time path. Specifically, `DGExplainer` first computes the relevance of the output for the last GRU cell's prediction. Then, at each time step $t$, `DGExplainer` derives the relevances of the inputs based on the relevance of the current cell's output. The inputs to each GRU cell are: 1) the GCN-encoded feature, and 2) the hidden state. The detailed process is shown below.

Given the final hidden state for a node, $R_{\mathbf{h}_T}$, where $\mathbf{h}_T = (\mathbf{H}_T^{(i,:)})^\top$, the objective is to compute the relevances of the inputs, $R_{\mathbf{h}_{t-1}}$ and $R_{\hat{\mathbf{x}}_{t-1}}$, from the relevance of the output, $R_{\mathbf{h}_t}$, for each GRU cell at time $t$. As

described in Section 3.1.2, relevance back-propagation redistributes the activation of a descendant neuron to its predecessor neurons, with the relevance being proportional to the weighted activation value. Based on the dependencies among different components in the final step of the GRU, as shown in Equation (2d), we derive the relevance back-propagation for this step as follows:

$$R_{\mathbf{h}_{t-1}} = R_{\mathbf{h}_{t-1}\leftarrow\mathbf{h}_t} + R_{\mathbf{h}_{t-1}\leftarrow\mathbf{n}} + R_{\mathbf{h}_{t-1}\leftarrow\mathbf{z}} + R_{\mathbf{h}_{t-1}\leftarrow\mathbf{r}}. \tag{7}$$

Note that neurons $\mathbf{r}$ and $\mathbf{z}$ only receive messages from neuron $\mathbf{h}_{t-1}$, as shown in Equations (2a) and (2b). Consequently, their contribution to $\mathbf{h}_t$ can be merged into the contribution from $\mathbf{h}_{t-1}$, and their relevances can be regarded as constants. Notice that $\mathbf{h}_{t-1}$ is used to compute both $\mathbf{n}$ in Equation (2c) and $\mathbf{h}_t$ in Equation (2d). This reveals that the relevance $R_{\mathbf{h}_{t-1}}$ has two sources: $\mathbf{n}$ and $\mathbf{h}_t$. Based on the contributions from $R_{\mathbf{h}_{t-1}\leftarrow\mathbf{n}}$ and $R_{\mathbf{h}_{t-1}\leftarrow\mathbf{h}_t}$, we can define $R_{\mathbf{h}_t}$ as follows:

$$R_{\mathbf{h}_t} = R_{\mathbf{h}_{t-1}} + R_{\mathbf{n}}, \tag{8}$$

Given that the relevance of a neuron is proportional to its activation at the same layer, i.e., $R_{k\leftarrow k_1} : R_{k\leftarrow k_2} = a_{k_1}^{(l)} : a_{k_2}^{(l)}$, we can derive the following based on Equation (2d):

$$\frac{R_{\mathbf{h}_{t-1}}}{R_{\mathbf{n}}} = \frac{a_{\mathbf{h}_{t-1}}}{a_{\mathbf{n}}} = \frac{\mathbf{z}\odot\mathbf{n}}{(1-\mathbf{z})\odot\mathbf{h}_{t-1}}. \tag{9}$$

We can conclude that if we derive $R_{\mathbf{h}_{t-1}\leftarrow\mathbf{h}_t}$ and $R_{\mathbf{h}_{t-1}\leftarrow\mathbf{n}}$, we can then obtain $R_{\mathbf{h}_t}$. Therefore, we break down this problem into three steps: computing $R_{\mathbf{h}_{t-1}\leftarrow\mathbf{h}_t}$, $R_{\mathbf{h}_{t-1}\leftarrow\mathbf{n}}$, and $R_{\mathbf{h}_{t-1}}$, as formulated below:

**(a) Compute $R_{\mathbf{h}_{t-1}\leftarrow\mathbf{h}_t}$:** Solving for Equations (8) and (9) obtains:

$$R_{\mathbf{h}_t\leftarrow\mathbf{n}} = \frac{\mathbf{z}\odot\mathbf{n}}{\mathbf{h}_t+\epsilon}\odot R_{\mathbf{h}_t}, \quad R_{\mathbf{h}_{t-1}\leftarrow\mathbf{h}_t} = \frac{(1-\mathbf{z})\odot\mathbf{h}_{t-1}}{\mathbf{h}_t+\epsilon}\odot R_{\mathbf{h}_t}, \tag{10}$$

where $\epsilon > 0$ is a constant introduced to keep the denominator non-zero. Notice that the only ancestor neuron of $\mathbf{n}$ is $\mathbf{h}_t$, so here $R_{\mathbf{n}\leftarrow\mathbf{h}_t}$ is actually $R_{\mathbf{n}}$, so in the following left of section, we use $R_{\mathbf{n}}$ for simplicity.

**(b) Compute $R_{\mathbf{h}_{t-1}\leftarrow\mathbf{n}}$:** From Equation (2c) we can calculate:

$$\mathbf{n}_1 := \mathbf{W}_{in}\hat{\mathbf{x}}_t, \quad \mathbf{n}_2 := \mathbf{r}\odot(\mathbf{W}_{hn}\mathbf{h}_{t-1}) = \mathbf{W}_{rn}\mathbf{h}_{t-1}, \quad \mathbf{b_n} := \mathbf{b}_{in} + \mathbf{r}\odot\mathbf{b}_{hn}. \tag{11a}$$

Then their relevance satisfies:

$$R_{\mathbf{n}} = R_{\mathbf{n}_1} + R_{\mathbf{n}_2} + R_{\mathbf{b}_n}, \quad R_{\mathbf{n}_1} : R_{\mathbf{n}_2} : R_{\mathbf{b}_n} = \mathbf{n}_1 : \mathbf{n}_2 : \mathbf{b}_n. \tag{12}$$

Hence, $R_{\mathbf{n}_1}$ and $R_{\mathbf{n}_2}$ can be obtained.

$$R_{\hat{\mathbf{x}}_t\leftarrow\mathbf{n}_1} = \sum_k \frac{\mathbf{W}_{in}^{(k,j)}\hat{\mathbf{x}}_t^{(j)}}{\epsilon+\sum_i \mathbf{W}_{in}^{(k,i)}\hat{\mathbf{x}}_t^{(i)}} R_{\mathbf{n}_1}^{(k)}. \tag{13}$$

Since $\mathbf{h}_{t-1}$ only influences $\mathbf{n}_2$ among the three parts of $\mathbf{n}$, we obtain $R_{\mathbf{h}_{t-1}\leftarrow\mathbf{n}}$ using $\epsilon$-rule for Equation (12):

$$R_{\mathbf{h}_{t-1}\leftarrow\mathbf{n}}^{(j)} = \sum_k \frac{\mathbf{W}_{rn}^{(k,j)}\mathbf{h}_{t-1}^{(j)}}{\epsilon+\sum_i \mathbf{W}_{rn}^{(k,i)}\mathbf{h}_{t-1}^{(i)}} R_{\mathbf{n}_2}^{(k)}. \tag{14}$$

**(c) Compute $R_{\mathbf{h}_{t-1}}$:** $R_{\mathbf{h}_{t-1}}$ can be computed by adding Equations (10) and (14) together: $R_{\mathbf{h}_{t-1}} = R_{\mathbf{h}_{t-1}\leftarrow\mathbf{h}_t} + \sum_j R_{\mathbf{h}_{t-1}\leftarrow\mathbf{n}}^{(j)}$. Notice that $R_{\hat{\mathbf{x}}_t}$ is the relevance of a node feature $\hat{\mathbf{x}}_t$, which is a row in $\hat{\mathbf{X}}_t$. By computing the set of relevances $\{R_{\hat{\mathbf{x}}_t^i}\}_{i=1}^N$ for all nodes, we can obtain the overall relevance matrix $R_{\hat{\mathbf{X}}_t}$, by concatenating the individual node relevances, i.e., $R_{\hat{\mathbf{X}}_t} = [R_{\hat{\mathbf{x}}_t^1}; R_{\hat{\mathbf{x}}_t^2}; \ldots; R_{\hat{\mathbf{x}}_t^N}]$.

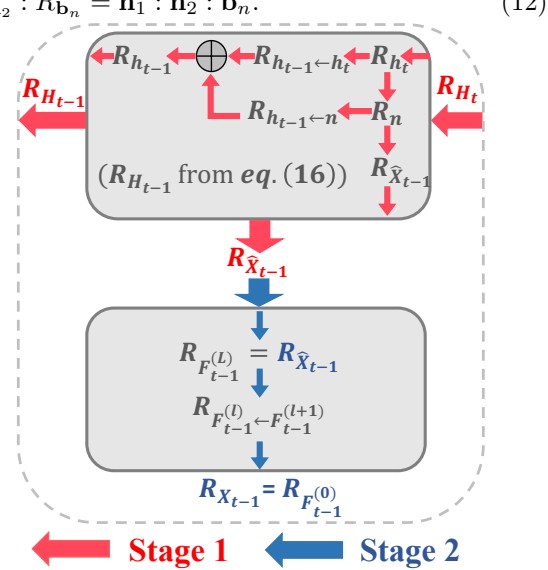

Figure 3: An illustration of `DGExplainer` for computing relevances in a backward manner. The feature relevance is computed by first back-propagating the prediction through the GRU and then the GCN.

### 3.2.2 Stage 2: Back-Propagate the Relevances in GCN

To find the relevance of the input data in a GCN, we start with the relevance of the output and backtrack through the network layers. We calculate the relevance of each layer's nodes using specific equations that distribute relevance from one layer to the previous. By repeating this process, we determine the relevance of the original input features. Finally, we average the absolute values of these relevances across all features to identify the importance of each node at a specific time. In the following, we show the concrete process:

Then we backtrack in the GCN to get $R_{\mathbf{X}_t}$ from $R_{\hat{\mathbf{X}}_t}$. Note that the $R_{\hat{\mathbf{X}}_t}$ is the relevance of the output $\hat{\mathbf{X}}$ of the GCN at the time step $t$ and $R_{\mathbf{F}_t^{(L)}} = R_{\hat{\mathbf{X}}_t}$. We can rewrite Equation (1) as:

$$\mathbf{F}_t^{(l+1)} = \sigma(\mathbf{P}_t^{(l)}\mathbf{W}_t^{(l)}); \quad \mathbf{P}_t^{(l)} := \mathbf{V}_t\mathbf{F}_t^{(l)}. \tag{15}$$

Let $(\mathbf{F}_t^{(l+1)})^{(k,:)}$, $(\mathbf{P}_t^{(l)})^{(k,:)}$, $(\mathbf{P}_t^{(l)})^{(:,k)}$, $(\mathbf{F}_t^{(l)})^{(:,k)}$ denote the $k$-th row of $\mathbf{F}_t^{(l+1)}$, the $k$-th row of $\mathbf{P}_t^{(l)}$, the $k$-th column of $\mathbf{P}_t^{(l)}$, the $k$-th column of $\mathbf{F}_t^{(l)}$, respectively. We have

$$(\mathbf{F}_t^{(l+1)})^{(k,:)} = \sigma((\mathbf{P}_t^{(l)})^{(k,:)}\mathbf{W}_t^{(l)}), \quad (\mathbf{P}_t^{(l)})^{(:,k)} := \mathbf{V}_t(\mathbf{F}_t^{(l)})^{(:,k)}. \tag{16}$$

Leveraging the $\epsilon$ rule, we assign the relevance by:

$$R_{(\mathbf{F}_t^{(l)})^{(j,k)}} = \sum_b \frac{\mathbf{V}^{(b,j)}(\mathbf{F}_t^{(l)})^{(j,k)}}{\epsilon + \sum_a \mathbf{V}_t^{(b,a)}(\mathbf{F}_t^{(l)})^{(a,k)}} R_{(\mathbf{P}_t^{(l)})^{(b,k)}}, \tag{17}$$

where $(\mathbf{W}_t^{(l)})^{(j,k)}$ represents the entry at the $j$-th row and $k$-th column of $\mathbf{W}_t^{(l)}$, and $\mathbf{V}_t^{(b,j)}$ denotes the entry at the $b$-th row and $j$-th column of $\mathbf{V}_t^{(k,j)}$. And the $R_{(\mathbf{P}_t^{(l)})^{(k,j)}}$ can be obtained similarly as $R_{(\mathbf{F}_t^{(l)})^{(j,k)}}$. The relevance $R_{\mathbf{F}_t^{(l)}}$ can be obtained from $R_{\mathbf{F}_t^{(l+1)}}$ using equation Equation (17), and $R_{(\mathbf{P}_t^{(l)})^{(k,j)}}$. Finally, the relevance $R_{\mathbf{F}_t^{(0)}}$ can be determined. Notice that $R_{\mathbf{F}_t^{(0)}} = R_{\mathbf{X}_t}$, so we have $R_{\mathbf{F}_t^{(0)}} = R_{\mathbf{X}_t}$, thus completing the backward process for obtaining relevance in the GCN. To further identify important nodes at a specific time step, we take the absolute values of the relevances and average them along the feature dimension to get the relevance of a node at time $t$: $R_{\mathbf{x}_t^i} = \sum_{j=1}^D |(R_{\mathbf{x}_t^i})^{(j)}|/D$.

---

**Algorithm 1** `DGExplainer`

---

**Input:** The input $\{\mathbf{X}_t\}_{t=1}^T$ and $\{\mathbf{A}_t\}_{t=1}^T$, the final relevance $\{R_{\mathbf{h}_T^j}\}_{j=1}^N$, the pre-trained model $f(\cdot)$.

**Output:** The relevances $\{R_{\mathbf{X}_t}\}_{t=1}^T$

1: // Forward process:
2: **for** each $t \in [1, T]$ **do**
3:      Compute $\hat{\mathbf{X}}_t$ via $\mathbf{F}_t^{(l+1)} = \sigma(\mathbf{V}_t\mathbf{F}_t^{(l)}\mathbf{W}_t^{(l)})$ with $\mathbf{F}_t^{(0)} = \mathbf{X}_t$, $\mathbf{F}_t^{(L)} = \hat{\mathbf{X}}_t$.
4:      **for** each $j \in [1, N]$ **do**
5:          Compute the hidden state $\mathbf{h}_t$ for the $j$-th sample
6:          $\hat{\mathbf{X}}_t^{(j,:)}$ via Equation (2) with $\mathbf{h}_{t-1}$.
7:      **end for**
8: **end for**
9: // Backward process:
10: **for** each $t = T, T-1, \ldots, 1$ **do**
11:      **for** each $j \in [1, N]$ **do**
12:          Compute $R_{\mathbf{n}}, R_{\mathbf{n_1}}, R_{\mathbf{n_2}}$ via Equation (12), $R_{\mathbf{h}_{t-1}}$ via Equations (10) and (14), and $R_{\hat{\mathbf{x}}_t}$ for the $j$-th sample $\hat{\mathbf{X}}_t^{(j,:)}$ via Equation (13) and hence obtain $R_{\hat{\mathbf{x}}_t^j}$.
13:      **end for**
14:      Stack $\{R_{\hat{\mathbf{x}}_t^j}\}_{j=1}^N$ to get $R_{\hat{\mathbf{X}}_t}$.
15:      Calculate $R_{\mathbf{X}_t}$ by iteratively applying Equation (17) with $R_{\hat{\mathbf{X}}_t} = R_{\mathbf{F}_t^{(L)}}$.
16: **end for**
17: **return** $\{R_{\mathbf{X}_t}\}_{t=1}^T$.

---

Table 1: Comparison with baseline methods in terms of fidelity ($\tau_1 = 0.8$), fidelity+ ($\tau_1 = 0.8$), sparsity ($\tau_2 = 3 \times 10^{-4}$), and stability ($r = 20\%$). The methods compared are GNNExplainer (GNNE), PGExplainer (PGE), SubgraphX (SubX), T-GNNExplainer (T-GNNE), and DyExplainer. 'Ours' refers to DGExplainer. For each metric, the best results are highlighted in **bold**, and the runner-up results are underlined.

| Dataset | Metric | SA | GNN-GI | GradCAM | GNNE | PGE | SubX | GCN-SE | T-GNNE | DyExplainer | Ours |
|---|---|---|---|---|---|---|---|---|---|---|---|
| Reddit | Fidelity ↑ | 0.35 | 0.34 | 0.33 | 0.29 | 0.28 | 0.24 | 0.32 | 0.39 | 0.35 | **0.42** |
| | Fidelity+ ↑ | 0.19 | 0.23 | 0.22 | 0.16 | 0.12 | 0.10 | 0.21 | 0.24 | **0.27** | **0.27** |
| | Sparsity ↑ | 0.79 | 0.86 | 0.53 | 0.67 | 0.75 | 0.34 | 0.71 | 0.86 | 0.84 | **0.87** |
| | Stability ↓ | 0.29 | 0.17 | 0.26 | 0.25 | 0.27 | 0.30 | 0.21 | 0.15 | 0.18 | **0.13** |
| PeMS04 | Fidelity ↑ | 0.30 | 0.29 | 0.26 | 0.24 | 0.19 | 0.18 | 0.33 | **0.44** | 0.37 | 0.39 |
| | Fidelity+ ↑ | 0.21 | 0.19 | 0.17 | 0.16 | 0.13 | 0.14 | 0.24 | 0.25 | **0.30** | 0.29 |
| | Sparsity ↑ | **0.99** | **0.99** | 0.95 | 0.92 | 0.90 | 0.87 | 0.91 | 0.97 | 0.98 | **0.99** |
| | Stability ↓ | 0.18 | 0.22 | 0.25 | 0.22 | 0.23 | 0.27 | 0.23 | 0.17 | 0.19 | **0.15** |
| PeMS08 | Fidelity ↑ | 0.26 | 0.25 | 0.20 | 0.19 | 0.15 | 0.13 | 0.26 | 0.27 | 0.28 | **0.30** |
| | Fidelity+ ↑ | 0.19 | 0.16 | 0.12 | 0.11 | 0.09 | 0.08 | 0.20 | 0.21 | **0.26** | 0.25 |
| | Sparsity ↑ | 0.94 | 0.94 | **0.95** | 0.91 | 0.92 | 0.90 | 0.92 | 0.94 | 0.94 | **0.95** |
| | Stability ↓ | 0.15 | 0.16 | 0.18 | 0.14 | 0.15 | 0.23 | 0.16 | 0.13 | 0.16 | **0.12** |
| Enron | Fidelity ↑ | 0.20 | 0.19 | 0.16 | 0.09 | 0.09 | 0.08 | 0.19 | 0.21 | 0.19 | **0.23** |
| | Fidelity+ ↑ | 0.14 | 0.15 | 0.11 | 0.06 | 0.07 | 0.05 | 0.13 | 0.15 | 0.17 | **0.18** |
| | Sparsity ↑ | 0.84 | 0.83 | 0.79 | 0.75 | 0.74 | 0.70 | 0.83 | 0.81 | 0.82 | **0.85** |
| | Stability ↓ | 0.13 | 0.15 | 0.17 | 0.15 | 0.16 | 0.19 | **0.11** | 0.19 | 0.17 | 0.15 |
| FB | Fidelity ↑ | 0.29 | 0.22 | 0.19 | 0.16 | 0.15 | 0.10 | 0.33 | 0.31 | 0.33 | **0.36** |
| | Fidelity+ ↑ | 0.18 | 0.14 | 0.13 | 0.11 | 0.09 | 0.07 | 0.17 | 0.20 | 0.22 | **0.23** |
| | Sparsity ↑ | 0.94 | 0.93 | 0.91 | 0.90 | 0.86 | 0.80 | 0.92 | **0.98** | 0.95 | 0.96 |
| | Stability ↓ | 0.13 | 0.15 | 0.17 | 0.16 | 0.14 | 0.18 | 0.22 | 0.16 | 0.18 | **0.12** |
| COLAB | Fidelity ↑ | 0.50 | 0.45 | 0.39 | 0.27 | 0.26 | 0.25 | 0.43 | **0.55** | 0.51 | 0.53 |
| | Fidelity+ ↑ | 0.32 | 0.30 | 0.25 | 0.19 | 0.18 | 0.20 | 0.28 | 0.33 | 0.29 | **0.35** |
| | Sparsity ↑ | 0.96 | 0.95 | 0.94 | 0.93 | 0.93 | 0.90 | 0.94 | **0.99** | 0.96 | 0.96 |
| | Stability ↓ | 0.18 | 0.25 | 0.27 | **0.16** | 0.19 | 0.25 | 0.24 | 0.21 | 0.24 | 0.18 |

Figure 3 illustrates the LRP process for a time step of these two states. Specifically, DGExplainer redistributes the relevance of the output hidden state, $R_{\mathbf{H}_t}$, to 1) the relevance of the input hidden state, $R_{\mathbf{H}_{t-1}}$, and 2) the relevance of the GCN-encoded feature, $R_{\hat{\mathbf{X}}_t}$. It then back-propagates the latter through the GCN and finally obtains the relevance of the input feature at this time step, $R_{\mathbf{X}_{t-1}^i}$. The entire algorithm is summarized in Algorithm 1.

## 4 Experiments

We conduct quantitative and qualitative experiments on six real-world graphs to address the following research questions:

- **RQ1**: Can the proposed DGExplainer learn high-quality explanations for the GCN-GRU model?

- **RQ2**: What are the benefits of DGExplainer in explaining dynamic GNNs compared to static methods?

- **RQ3**: How do the hyperparameters affect DGExplainer?

Unless otherwise specified, we present the performance of DGExplainer on the GCN-GRU model in our experiments. Additionally, in Appendix A.5, we demonstrate the performance of DGExplainer across various other dynamic GNN models.

### 4.1 Experiment Settings

**Datasets.** We evaluate the proposed framework on six real-world datasets. For the link prediction tasks, we use four datasets: Reddit Hyperlink (Reddit) (Kumar et al., 2018), Enron (Klimt & Yang, 2004), Facebook (FB) (Trivedi et al., 2019), and COLAB (Rahman & Al Hasan, 2016). For the node regression tasks, we use two datasets: PeMS04 and PeMS08 (Guo et al., 2019)[1]. The statistics of these datasets and the initial performance of GCN-GRU on them are presented in Appendix A.1.

**Baselines.** We assess our proposed method against eight baseline explanation methods. These include two general explanation methods: (a) Sensitivity Analysis (SA) (Baldassarre & Azizpour, 2019) and (b) GradCAM (Pope et al., 2019). Additionally, we compare our method with six GNN explanation methods: (c) GNN-GI (Schnake et al., 2020), (d) GNNExplainer, (e) PGExplainer, (f) SubgraphX, (g) GCN-SE, (h) T-GNNExplainer, and (i) DyExplainer. Detailed descriptions of these baseline methods are provided in Appendix A.2.

To Revwier wvjk: Weakness 1

**Evaluation.** We compare the quality of each explanation baseline and our proposed method using four quantitative metrics: confidence, sparsity, stability, and fidelity. Details of these evaluation metrics are elaborated in Appendix A.3. Following the experimental setup of a previous work (Pareja et al., 2020), we conduct experiments on link prediction and node classification.

- **Link prediction**: For this task, we concatenate the feature embeddings of nodes $u$ and $v$ as $[(\mathbf{h}_T^u)^\top; (\mathbf{h}_T^v)^\top]^\top$ and use a multi-layer perceptron (MLP) to predict the link probability by optimizing the cross-entropy loss. We experiment with the Reddit, Enron, FB, and COLAB datasets and use the Area Under the Curve (AUC) as the evaluation metric. Following a previous study (Pareja et al., 2020), in the implementation of the MLP, we use a rectified linear unit (ReLU) as the activation function for all layers except the output layer, where we apply the softmax function.

To Revwier WFn7: Further question about $\epsilon$

- **Node regression**: To predict the value for a node $u$ at time $t$, we apply a softmax activation function to the last layer of the GCN, resulting in the probability vector $\mathbf{h}_t^u$. We use the PeMS04 and PeMS08 datasets for this task and evaluate the performance using the mean absolute error (MAE) metric.

**Implementation Details.** We conducted all our experiments on a Linux machine equipped with four NVIDIA RTX A4000 Ti GPUs, each with 16GB of RAM. We used a two-layer GCN and trained the model for 1000 epochs using the Adam optimizer (Kingma & Ba, 2014), with an initial learning rate of 0.01. We use the ReLU as the activation function for all layers, except for the output layer for the GCN-GRU model, where the softmax function is applied. For the link prediction task, we employed a two-layer MLP with 64 hidden units. We tested the stabilizer $\epsilon$ with values $\{1e\text{-}5, 1e\text{-}4, 1e\text{-}3, 1e\text{-}2, 1e\text{-}1, 1, 2\}$, and finally choose $\epsilon = 0.1$ to implement the proposed method. In stability experiments, we set $r$ to $\{5\%, 10\%, 15\%, 20\%, 30\%\}$. The model performance results are based on the average analysis of 10 runs. The output embedding of a node $u$ produced by the GCN-GRU model at time $t$ is represented by $\mathbf{h}_t^u$.

To Revwier WFn7: Further question about $\epsilon$

### 4.2 Prediction and Explanation Performance

To address **RQ1**, we conducted a comprehensive comparison of our proposed method, `DGExplainer`, against several baseline methods. Our evaluation focused on two key aspects: prediction accuracy and the quality of explanations in identifying important nodes. The results demonstrate that `DGExplainer` outperforms the baselines in terms of fidelity and sparsity, providing more accurate and concise explanations. Additionally, our method exhibits good stability, ensuring consistent explanations even in the presence of minor perturbations, although on some datasets it slightly underperforms SA and GradCAM. These results establish the effectiveness and reliability of our proposed method in capturing important nodes and providing reliable explanations in the context of link prediction and node regression tasks.

**Results on fidelity, fidelity+ and sparsity.** Fidelity measures a method's ability to accurately capture important nodes. A high-fidelity explanation method is desirable. Fidelity+ is a surrogate version of fidelity, where a graph is sampled from the explanation subgraph by retaining each edge with probability $\alpha$ and erasing it with probability $1 - \alpha$. To assess fidelity, we ranked the nodes based on their importance and

---

[1] pems.dot.ca.gov

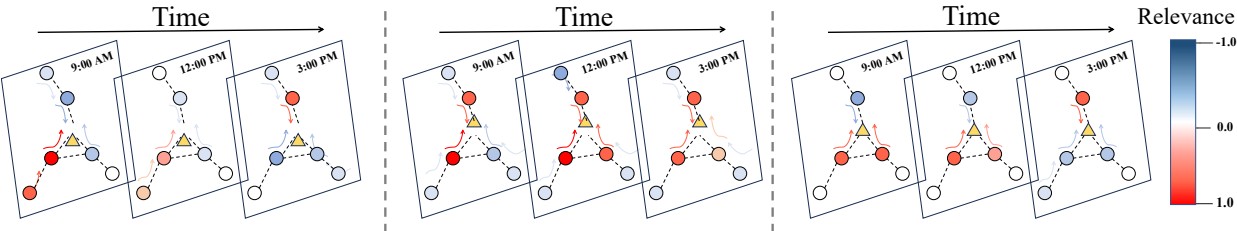

Figure 4: Illustration of the proposed method applied to the PeMS04 dataset. In this figure, warm colors indicate positive effects, while cold colors denote negative effects. The intensity of the color corresponds to the magnitude of the effect. From left to right, the subfigures represent the visualization results of GNN-GI, GNNExplainer, and the proposed method.

conducted occlusion experiments by selectively occluding a fraction of the top nodes while keeping 80% of the nodes unchanged ($\tau_1 = 0.8$). For fidelity+, we define the explanation subgraph as the subgraph consisting of nodes with relevance greater than $\tau_1$. The proposed method consistently outperformed the baselines in terms of both fidelity, fidelity+, and sparsity across most datasets, as shown in Table 1. In the remaining datasets, our method achieved comparable results. The Fidelity+ gap between the proposed method and the baselines is larger than the Fidelity gap, further demonstrating DGExplainer's effectiveness in assigning higher relevance to important nodes.

To Revwier 1zv7: Weakness 1

**Results on stability.** A stability evaluation was conducted to assess how well the explanation method handles perturbations in the input graph. We introduced random perturbations by adding additional edges to the original graph at a ratio of $r = 20\%$ and evaluated the resulting changes in the relevances generated by the model. A stable explanation method should provide consistent explanations when the input undergoes minor perturbations, resulting in lower stability scores. As presented in Table 1, our proposed method generally exhibited good stability, although it did not outperform SA and GNNExplainer. These findings indicate that our method demonstrates relative robustness to small perturbations in the input graph.

To Revwier wvjk: Weakness 2

### 4.3 Qualitative Analysis

To address **RQ2**, we conducted quantitative experiments and visualizations of the generated explanations using `DGExplainer` and baseline methods on the PeMS04 dataset, which represents traffic flow on a highway network. The results, presented in Figure 4, indicate that `DGExplainer` generates the most reasonable and detailed explanations compared to the GNN-GI and GNNExplainer approaches. Our analysis revealed several key findings: (a) GNN-GI tends to assign equally extreme relevances to every individual node, suggesting that each node has a strong correlation with the prediction. In contrast, GNNExplainer generates average scores for all the identified nodes. (b) GNN-GI identifies nearly all nodes as important, while GNNExplainer only identifies a few nodes as significant, disregarding the correlations of other nodes with the target variable.

These disparities in the visualization results are due to the fact that the comparison methods fail to capture the temporal patterns of dynamic graphs, treating each time step independently and considering only spatial information. In contrast, `DGExplainer` excels in generating comprehensive and context-aware explanations by effectively incorporating temporal dynamics into the analysis. By considering both spatial and temporal information, `DGExplainer` provides a more accurate understanding of the underlying relationships within the dynamic GNNs.

### 4.4 Parameter Sensitivity Analysis

To address **(RQ3)**, we investigate fidelity across various threshold values, denoted as $\tau_1 = \{0.5, 0.6, 0.7, 0.8, 0.9\}$. The fidelity analysis is presented in Figure 5. Our observations are as follows: (a) With smaller $\tau_1$ values, the fidelity is high. This is because a larger number of nodes are occluded when their relevance surpasses the threshold, resulting in a substantial change in accuracy. (b) As $\tau_1$ increases, the fidelity gradually decreases, with a steeper decline observed in the range of $[0.8, 0.9]$. Overall, our proposed method consistently achieves the highest fidelity across all thresholds and datasets, affirming the robustness

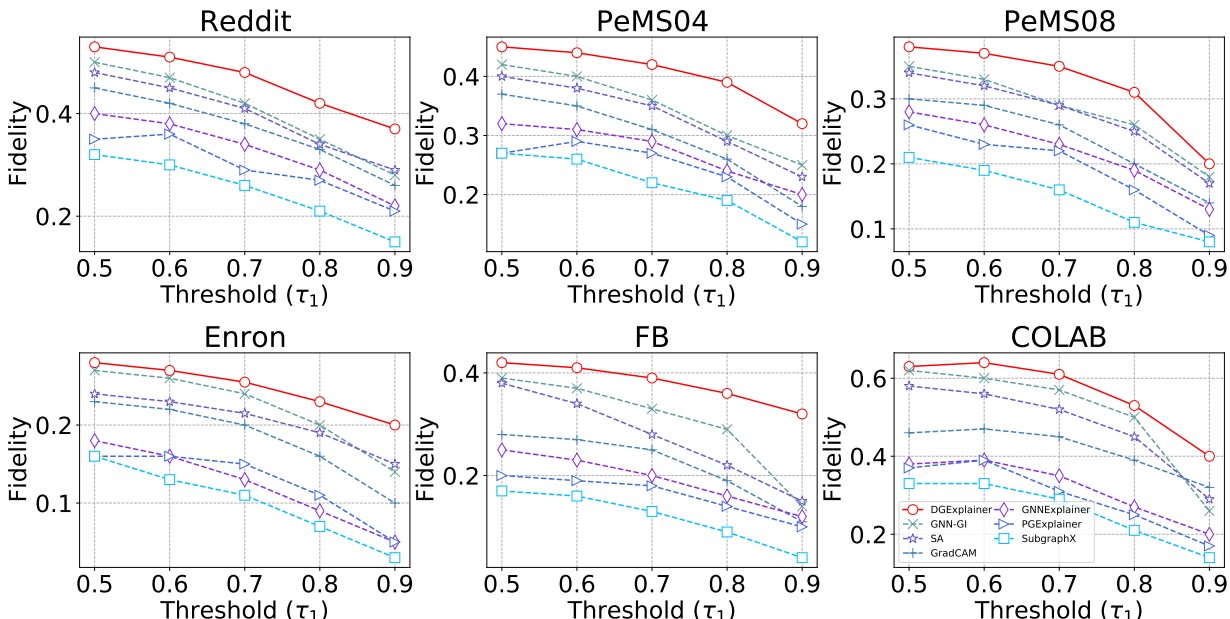

Figure 5: Comparison of different methods with the fidelity of similar levels of thresholds.

of our framework. These findings provide substantial insights into the relationship between fidelity and the chosen threshold values, reinforcing the efficacy of our approach.

## 4.5 Related Work

We review previous studies related to our work, focusing first on the recent advances in dynamic graph neural networks and then on existing explainability methods for static GNNs.

### 4.5.1 Dynamic Graph Neural Networks

Dynamic graph neural networks (Dynamic GNNs) consider both temporal and graph-structural information to tackle dynamic graphs. These networks are commonly applied in social media, citation networks, transportation networks, and pandemic networks. DANE (Li et al., 2017) is an efficient dynamic GNN that updates node embeddings using the eigenvectors of the graph's Laplacian matrix, based on the graph from the previous time step. CTDANE (Nguyen et al.) and NetWalk (Yu et al., 2018b) extend random walk-based approaches by enforcing temporal rules on the walks. Additionally, embedding methods aggregate neighboring node features. For example, DynGEM (Goyal et al., 2018) and Dyngraph2vec (Goyal et al., 2020) use deep autoencoders to encode snapshots of dynamic graphs.

A prevalent category of approaches combines GNNs with recurrent architectures, where the GNNs extract graph-structural information and the recurrent units handle sequential flows. GCRN (Seo et al., 2018) leverages GCN layers to obtain node embeddings and feeds them into recurrent layers to track dynamism. STGCN (Yu et al., 2018a), which stacks ST-Conv blocks, proposes a sophisticated architecture that effectively captures complex localized spatial-temporal correlations. Instead of directly integrating RNNs into the entire structure, EvolveGCN (Pareja et al., 2020) uses RNNs to update the weights of GCNs. Another approach (Hajiramezanali et al., 2019) introduces variational autoencoder versions for dynamic graphs, VGRNN and SI-VGRNN. Both models use a GCN integrated into an RNN as an encoder to track the temporal evolution of the graph. The GCN-GRU model used to demonstrate the proposed method has wide applications (Gui et al., 2020; Yang et al., 2020; Zhao et al., 2018). The two modules are co-trained to capture the spatial-temporal information in dynamic graphs. For example, in traffic flow prediction, the GCN models the dynamics of traffic as an information dissemination process. Meanwhile, the GRU captures

dependencies across different time steps through gate units that are trained to manage inputs and memory states, enabling the retention of information over longer periods (Zhao et al., 2018).

To Revwier 1zv7: More Dynamic GNN related work.

### 4.5.2 Explainability on GNNs

Existing explainability approaches for GNNs are mostly focused on static GNNs and can be categorized into four main directions. The first direction focuses on highlighting the importance of various input features by analyzing feature gradients and relevances. For instance, Sensitivity Analysis (Baldassarre & Azizpour, 2019) assigns importance scores to different input features using the squared values of their gradients. GNN-GI employs the Grad⊙Input method (Shrikumar et al., 2016; Sanchez-Lengeling et al., 2020), which calculates feature contribution scores as the element-wise product of the input features. GNN-LRP (Schnake et al., 2021) assigns a relevance score to each walk, representing a message flow path within the graph. This relevance score is determined using the T-order Taylor expansion of the model with respect to the incorporation operator. A recent study (Xiong et al., 2023) introduces two novel relevant walk search algorithms based on max-product message passing, reducing the computational complexity of GNN-LRP from exponential to polynomial time. Additionally, GradCAM (Pope et al., 2019) adapts the Class Activation Mapping (CAM) approach (Pope et al., 2019) for general graph classification models by removing the requirement for a global average pooling layer.

To Revwier 1zv7: Minor suggestion

The second research direction focuses on accumulating local effects to create a locally faithful approximation. For example, GraphLime (Huang et al., 2022) utilizes neighboring nodes as perturbed inputs and employs a nonlinear surrogate model. This model is capable of assigning significant weights to features that are crucial for inference. In addition, PGM-Explainer (Vu & Thai, 2020) leverages an interpretable Bayesian network to approximate the predictions needed for explanation.

The third line of research examines perturbation-based interpretation methods, which identify key components affecting model predictions by perturbing nodes, edges, or features. Specifically, GNNExplainer (Ying et al., 2019) and PGExplainer (Luo et al., 2020) identify key features by maximizing mutual information between the perturbed and input graphs. Graphsvx (Duval & Malliaros, 2021) and SubgraphX (Yuan et al., 2021) use the Shapley value (Shapley, 1953) to assess the importance of features and nodes in Graphsvx and subgraphs in SubgraphX. TempME (Chen & Ying, 2023) samples temporal motifs around the interaction between two nodes' predictions, assigning importance scores while balancing explanation accuracy and information compression by maximizing mutual information with the target prediction and minimizing it with the original temporal graph. SubMT (Chen et al., 2024) introduces an interpretable subgraph learning method that identifies the subgraph multilinear extension as a factorized distribution. The extracted subgraph is considered interpretable and faithful when its prediction strongly correlates with the subgraph's sampling probability. GraphMask (Schlichtkrull et al., 2020) trains a classifier to generate edge masks for each GNN layer, identifying edges that can be removed without altering the model's predictions.

To Revwier 1zv7: Minor suggestion

The final direction focuses on model-level explanations, offering a broader understanding of how the model operates as a whole. This method provides general insights into the overall functioning of the model, rather than focusing on individual predictions. For example, XGNN (Yuan et al., 2020) offers a global explanation of GNNs by training a graph generator to create patterns that maximize the predictions of a given model.

To Revwier 1zv7: Weakness 1

Despite the success of existing explainability methods, they primarily focus on static GNNs and cannot be directly applied to explain dynamic GNNs because they overlook the temporal or dynamic aspects of graphs. Recently a dynamic GNN explainer, DyExplainer (Wang et al., 2023), has been proposed. Specifically, DyExplainer trains a dynamic GNN to produce node embeddings and uses structural and temporal attention by capturing both pivotal relationships within snapshots and temporal dependencies across long-term snapshots. While explaining dynamic GNNs remains a critical challenge, only a limited number of approaches have been proposed to address this issue. This motivates us to fill this gap by proposing our method.

To Revwier wvjk: Weakness 4

## 5 Conclusion

In this paper, we present `DGExplainer`, a novel and efficient framework that utilizes both layer-wise and time-wise relevance back-propagation to explain the predictions of dynamic Graph Neural Networks (GNNs).

To evaluate `DGExplainer`'s performance, we conduct both quantitative and qualitative experiments. The results demonstrate the framework's effectiveness in identifying crucial nodes for link prediction and node regression tasks, outperforming existing explanation methods. This research pioneers the exploration of dynamic GNNs, offering insights into their intricate structures, which is a significant challenge due to the complexity of inference in time-varying modules. Unlike existing static GNN explainers, `DGExplainer` does not require learning a surrogate function or executing any optimization procedures. Additionally, it can be extended to other advanced dynamic GNNs.

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

## A    Appendix

In this appendix, we provide a more detailed introduction to the datasets, describe the evaluation metrics for baselines, explain the LRP method in detail, and show additional experiments to demonstrate the superiority of the proposed method, `DGExplainer`.

### A.1    Datasets

The statistics of the datasets and the initial performance of GCN-GRU on these datasets are summarized in Table 2.

Table 2: Dataset statistics and performance metrics of the GCN-GRU model. We report the AUC (%) for the Reddit, Enron, FB, and COLAB datasets, and the MAE for the PeMS04 and PeMS08 datasets.

| Dataset | Reddit | PeMS04 | PeMS08 | Enron | FB | COLAB |
|---|---|---|---|---|---|---|
| # Nodes | 55,863 | 307 | 170 | 184 | 663 | 315 |
| # Edges | 858,490 | 680 | 340 | 266 | 1068 | 308 |
| # Train/Test | 122/34 | 45/14 | 50/12 | 8/3 | 6/3 | 7/3 |
| # Time Step | 6 | 4 | 4 | 4 | 4 | 4 |
| Performance | 0.702 | 55.29 | 59.35 | 0.951 | 0.870 | 0.879 |

- **Reddit** is a directed network extracted from posts that generate hyperlinks connecting one subreddit to another. It includes various features, such as the source post, target URL, post title, and comment text, along with metadata like the number of upvotes and downvotes each post and comment received. The Reddit Hyperlink dataset comprises hyperlink information from over 3 million posts and their associated comments on the social media platform Reddit, spanning from 2008 to 2016.

- **PeMS04** and **PeMS08** are real-time traffic flow datasets providing traffic information for the state of California, USA. The PeMS04 dataset includes traffic flow data from over 39,000 sensors, while the PeMS08 dataset includes data from over 40,000 sensors. These sensors are located on freeways and arterial roads throughout California. The datasets cover the periods from January 1, 2018, to December 31, 2018, and from January 1, 2020, to December 31, 2020, respectively. Both datasets are collected at 5-minute intervals and include information on traffic speed, occupancy, and volume, resulting in 288 data points per detector per day. Additionally, the datasets include weather information and incident reports, which can be used to analyze the impact of weather and incidents on traffic flow. The data are transformed using zero-mean normalization to ensure the average is 0.

- **Enron**, **FB**, and **COLAB**: These datasets are dynamic graphs constructed from different types of interactions: email messages exchanged between employees, co-author relationships among authors, and Facebook wall posts, respectively. The Enron dataset represents the email communication network of employees at the Enron Corporation, where nodes represent individuals and edges represent email messages sent between them over time. The FB dataset captures the social network of Facebook users, where nodes represent users and edges represent friendship connections. Finally, the COLAB dataset contains transcripts of meetings held by community organizations, where nodes represent participants and edges represent their interactions during the meetings. We collected and processed these three datasets following the methodology described in (Hajiramezanali et al., 2019).

## A.2 Baselines

We provide detailed descriptions of the baselines used for comparison in our experiments in the follows:

- (a) **Sensitivity Analysis (SA)** (Baldassarre & Azizpour, 2019) computes importance scores using squared gradients of input features through back-propagation. It assumes that higher absolute gradient values indicate greater importance, but it fails to accurately represent importance and is prone to saturation issues (Shrikumar et al., 2017).

- (b) **GradCAM** (Selvaraju et al., 2017) extends the Class Activation Mapping (CAM) (Zhang et al., 2018) method to graph classification by removing the global average pooling layer constraint and mapping the final node embeddings to the input space for measuring node importance. It uses gradients as weights to combine different feature maps, computed by averaging the gradients of the target prediction with respect to the final node embeddings.

- (c) **GNN-GI** (Schnake et al., 2021) adopts Grad⊙Input (GI) (Shrikumar et al., 2017), which quantifies the contribution of features by computing the element-wise product of the input features and the gradients of the decision function with respect to those features. As a result, GI takes into account both the sensitivity of features and the scale of their values.

- (d) **GNNExplainer** (Ying et al., 2019) generates explanations for predictions in the form of subgraphs and feature masks that highlight the relevant parts of the input data. It provides explanations by generating a compact subgraph from the input graph, along with a select subset of node features that greatly influence the prediction.

- (e) **PGExplainer** (Luo et al., 2020) leverages a deep neural network parameterized explainer to generate global explanations that highlight important subgraphs influencing a model's predictions. This method endows PGExplainer with a natural capacity to deliver multi-instance explanations.

- (f) **SubgraphX** (Yuan et al., 2021) identifies important subgraphs measured by Shapley values. It employs the Monte Carlo tree search algorithm for efficiently exploring various subgraphs within a given input graph.

- (g) **GCN-SE** (Fan et al., 2021) computes the importance of different graph snapshots by measuring the change in accuracy after masking the attention in that timestep.

- (h) **T-GNNExplainer** (Xia et al., 2022) finds a subset of historical events that lead to the prediction, given a temporal prediction of a model. This method regards a temporal graph as a sequence of temporal events between nodes.

## A.3 Evaluation Metrics

We present a comprehensive overview of the four key quantitative metrics that have been instrumental in our analysis: confidence, sparsity, stability, and fidelity. The subsequent sections provide a detailed exposition of each metric.

- **Fidelity** characterizes whether the explanations are faithfully important to the model predictions (Sanchez-Lengeling et al., 2020). In the experiment, we measure fidelity by calculating the difference in classification accuracy or regression errors obtained by occluding all nodes with importance

values greater than a threshold $\tau_1$ on a scale of $(0, 1)$. We averaged the fidelity across classes for each method. This approach is equivalent to the Fidelity+ metric proposed by Zheng et al. (2023).

- **Surrogate Fidelity** (Zheng et al., 2023) addresses the Out Of Distribution (OOD) problem in the resultant subgraphs caused by removing edges from the original graph and introduces surrogate fidelity metrics to mitigate this issue. Specifically, for the fidelity+ metric defined earlier, the surrogate version employs a function $E_\alpha$ that stochastically samples a graph by retaining each edge with probability $\alpha$ and erasing it with probability $1 - \alpha$. It has been shown in practice that choosing $\alpha < 1$ provides a more appropriate fidelity measure, as this approach helps to alleviate the OOD problem.

- **Sparsity** measures the fraction of nodes selected for an explanation (Yuan et al., 2021; Pope et al., 2019). It evaluates whether the model efficiently marks the most contributive part of the dataset. High sparsity scores indicate that fewer nodes are identified as important. In our experiment, we compute sparsity by calculating the ratio of nodes with saliency values or relevances lower than a predefined threshold $\tau_2$ on a scale of $(0, 1)$.

- **Stability** assesses the consistency of explanations when small changes are applied to the input (Sanchez-Lengeling et al., 2020). Good explanations should be stable, meaning they remain approximately the same under small input perturbations. To evaluate stability, we randomly add more edges at a ratio of $r\%$ and measure the change in relevances/importances produced by the model.

### A.4 More Details About Layer-wise Relevance Propagation

Layer-wise relevance propagation (LRP) (Bach et al., 2015) is a technique for explaining the predictions of deep neural networks. It operates on the assumption that a neuron's relevance is proportional to its weighted activation value. This follows the intuition that a larger output activation indicates that the neuron carries more information and contributes more significantly to the final result.

The concept behind LRP assumes that the relevance, denoted as $R_{k_2}^{(l+1)}$, is known for a neuron in the subsequent layer $(l + 1)$. This assumption allows us to break down and distribute this relevance to the neurons, denoted as $k_1$, in the current layer $l$ that contribute input to the neuron $k_2$. This process enables us to determine the relevance value for a neuron $k_1$ in layer $l$ by aggregating all the incoming messages from neurons in layer $(l + 1)$. A notable challenge in LRP is formulating an appropriate rule for redistributing relevance across each layer. Drawing insights from prior studies (Bach et al., 2015; Binder et al., 2016; Schnake et al., 2021), we describe the propagation rule as follows:

$$R_{k_1 \leftarrow k_2}^{(l,l+1)} = \sum_{k_2} \frac{\mathbf{W}_{k_1 k_2} a_{k_1}^{(l)}}{\epsilon + \sum_{k_1} \mathbf{W}_{k_1 k_2} a_{k_1}^{(l)}} R_{k_2}^{(l+1)}, \tag{18}$$

where $\mathbf{W}_{k_1 k_2}$ represents the connection weight between neurons $k_1$ and $k_2$. $R_{k_2}^{(l+1)}$ is the relevance for neuron $k_2$ at layer $(l + 1)$, and $R_{k_1 \leftarrow k_2}^{(l,l+1)}$ is the relevance for neuron $k_1$ derived from $k_2$ at layer $l$. $a_{k_1}^{(l)}$ denotes the activation of neuron $k_1$ at layer $l$. The term $\epsilon$ is a predefined stabilizer that prevents the denominator from being zero. Clearly, the connection between the relevance and the weighted activation $\mathbf{W}_{k_1 k_2} a_{k_1}^{(l)}$ is discernible. This relationship indicates that the relevance varies in proportion to the magnitude of the weighted activation. Additionally, the nature of the contribution, whether positive or negative, depends on the sign of the weighted activation. In our implementation, we use the softmax activation function, ensuring that the weighted activations are in the range $(0, 1)$. As a result, the denominator will always be non-zero after adding $\epsilon$. .

Epsilon Rule (LRP-$\epsilon$) (Bach et al., 2015). A first enhancement of the basic LRP-0 rule consists of adding a small positive term $\epsilon$ in the denominator: The work in (Bach et al., 2015) established two formulas for computing the messages $R_{k_1 \leftarrow k_2}^{(l,l+1)}$. The first formula called $\epsilon$-rule is given by

$$R_{k_1 \leftarrow k_2}^{(l,l+1)} = \frac{z_{k_1 k_2}}{z_{k_2} + \epsilon \cdot \text{sign}(z_{k_2})} R_{k_2}^{(l+1)}, \tag{19}$$

with $z_{ij} = (w_{ij} x_i)^p$ and $z_j = \sum_{k : w_{kj} \neq 0} z_{kj}$. The variable $\epsilon$ is a stabilizer term whose purpose is to avoid numerical degenerations when $z_j$ is close to zero, and which is chosen to be small.

Epsilon Rule (LRP-$\epsilon$) (Bach et al., 2015). A first enhancement of the basic LRP-0 rule consists of adding a small positive term $\epsilon$ in the denominator:

$$R_{k_1 \leftarrow k_2}^{(l,l+1)} = \sum_{k_2} \frac{a_{k_1}^{(l)} \mathbf{W}_{k_1 k_2}}{\epsilon + \sum_{k_2,k_1} a_{k_1}^{(l)} \mathbf{W}_{k_1 k_2}} R_{k_2}^{(l+1)}$$

The role of $\epsilon$ is to absorb some relevance when the contributions to the activation of neuron $k$ are weak or contradictory. As $\epsilon$ becomes larger, only the most salient explanation factors survive the absorption. This typically leads to explanations that are sparser in terms of input features and less noisy.

Therefore, by summing up the relevance over all neurons $k_2$ in layer $(l + 1)$, based on Equation (19). The Equation (18) can be obtained from Equation (19).

$$R_{k_1 \leftarrow k_2}^{(l,l+1)} = \sum_{k_2} \frac{\mathbf{W}_{k_1 k_2} a_{k_1}^{(l)}}{\epsilon + \sum_{k} \mathbf{W}_{k k_2} a_{k}^{(l)}} R_{k_2}^{(l+1)}.$$

## A.5 More Experiments on Dynamic GNN Architectures

We conducted additional experiments on diverse dynamic GNN architectures, including, Evolve-GCN (Pareja et al., 2020), DySAT (Sankar et al., 2018), GC-LSTM (Chen et al., 2022), and ROLAND (You et al., 2022).

Table 3: Experimental results on other dynamic GNNs, in terms of fidelity ($\tau_1 = 0.8$), sparsity ($\tau_2 = 3 \times 10^{-4}$), and stability ($r = 20\%$). For each metric, the best results are highlighted in **bold** text.

| | | Evolve-GCN | | | DySAT | | | GC-LSTM | | | ROLAND | | |
|---|---|---|---|---|---|---|---|---|---|---|---|---|---|
| | | T-GNNE | DyExplainer | Ours | T-GNNE | DyExplainer | Ours | T-GNNE | DyExplainer | Ours | T-GNNE | DyExplainer | Ours |
| Reddit | Fidelity ↑ | 0.27 | 0.30 | **0.32** | 0.22 | 0.28 | **0.31** | 0.18 | 0.26 | **0.27** | 0.33 | **0.43** | 0.42 |
| | Sparsity ↑ | 0.74 | 0.86 | **0.88** | 0.72 | 0.84 | **0.85** | 0.78 | **0.91** | 0.90 | 0.75 | 0.80 | **0.81** |
| | Stability ↓ | 0.25 | 0.19 | **0.14** | 0.23 | 0.16 | **0.15** | 0.22 | 0.20 | **0.18** | 0.27 | 0.25 | **0.21** |
| PeMS04 | Fidelity ↑ | 0.35 | 0.41 | **0.43** | 0.15 | **0.23** | 0.22 | 0.33 | 0.35 | **0.36** | 0.27 | **0.30** | **0.30** |
| | Sparsity ↑ | **0.99** | 0.95 | **0.99** | 0.96 | 0.92 | **0.99** | 0.94 | 0.95 | **0.99** | 0.97 | **0.99** | **0.99** |
| | Stability ↓ | 0.14 | 0.16 | **0.11** | 0.33 | 0.35 | **0.30** | 0.30 | 0.29 | **0.27** | **0.19** | 0.25 | 0.23 |
| PeMS08 | Fidelity ↑ | 0.21 | 0.26 | **0.31** | 0.15 | 0.20 | **0.22** | 0.16 | **0.21** | **0.21** | 0.28 | 0.31 | **0.32** |
| | Sparsity ↑ | **0.98** | 0.94 | 0.95 | 0.87 | 0.86 | **0.90** | 0.88 | 0.89 | **0.91** | 0.85 | **0.91** | 0.90 |
| | Stability ↓ | 0.18 | 0.22 | **0.16** | 0.21 | 0.20 | **0.17** | 0.19 | 0.20 | **0.15** | 0.18 | 0.14 | **0.11** |
| Enron | Fidelity ↑ | **0.21** | 0.23 | 0.24 | **0.17** | 0.19 | 0.20 | **0.22** | 0.25 | 0.24 | **0.24** | 0.25 | 0.27 |
| | Sparsity ↑ | 0.78 | 0.83 | **0.87** | 0.80 | 0.79 | **0.83** | 0.76 | 0.81 | **0.82** | 0.70 | 0.76 | **0.79** |
| | Stability ↓ | 0.24 | 0.22 | **0.16** | 0.22 | 0.19 | **0.11** | 0.20 | 0.23 | **0.19** | 0.11 | 0.15 | **0.08** |
| FB | Fidelity ↑ | 0.23 | 0.27 | **0.33** | 0.33 | 0.34 | **0.37** | 0.19 | 0.21 | **0.23** | 0.09 | **0.13** | 0.11 |
| | Sparsity ↑ | 0.89 | 0.95 | **0.96** | 0.77 | 0.82 | **0.87** | 0.86 | 0.91 | **0.94** | 0.88 | 0.89 | **0.92** |
| | Stability ↓ | 0.15 | 0.13 | **0.11** | 0.21 | 0.19 | **0.17** | 0.25 | 0.23 | **0.19** | 0.19 | **0.14** | 0.15 |
| COLAB | Fidelity ↑ | 0.39 | 0.43 | **0.47** | 0.38 | **0.44** | 0.43 | 0.37 | 0.39 | **0.41** | 0.32 | 0.36 | **0.38** |
| | Sparsity ↑ | 0.94 | 0.97 | **0.98** | 0.92 | 0.94 | **0.97** | 0.91 | **0.99** | 0.98 | 0.92 | 0.97 | **0.99** |
| | Stability ↓ | 0.19 | 0.15 | **0.11** | 0.21 | 0.19 | **0.17** | 0.22 | **0.19** | **0.19** | **0.14** | 0.16 | 0.15 |

From the results in Table 3, we can observe that Evolve-GCN generally offers higher fidelity and is easier to explain compared to other architectures. In contrast, other dynamic GNNs, such as ROLAND, show varying levels of explainability difficulty; for instance, ROLAND is particularly challenging to explain on the FB dataset. Overall, while other methods may produce more accurate and stable explanations, they do so with higher fidelity and sparsity, but at the cost of lower stability.

To Revwier 1zv7: Weakness 2

## A.6 Ablation Study

To demonstrate the importance of capturing temporal dependencies in explaining dynamic graphs, we perform an ablation study by treating the hidden representation $H_t$ as the output of the GNN, denoted by

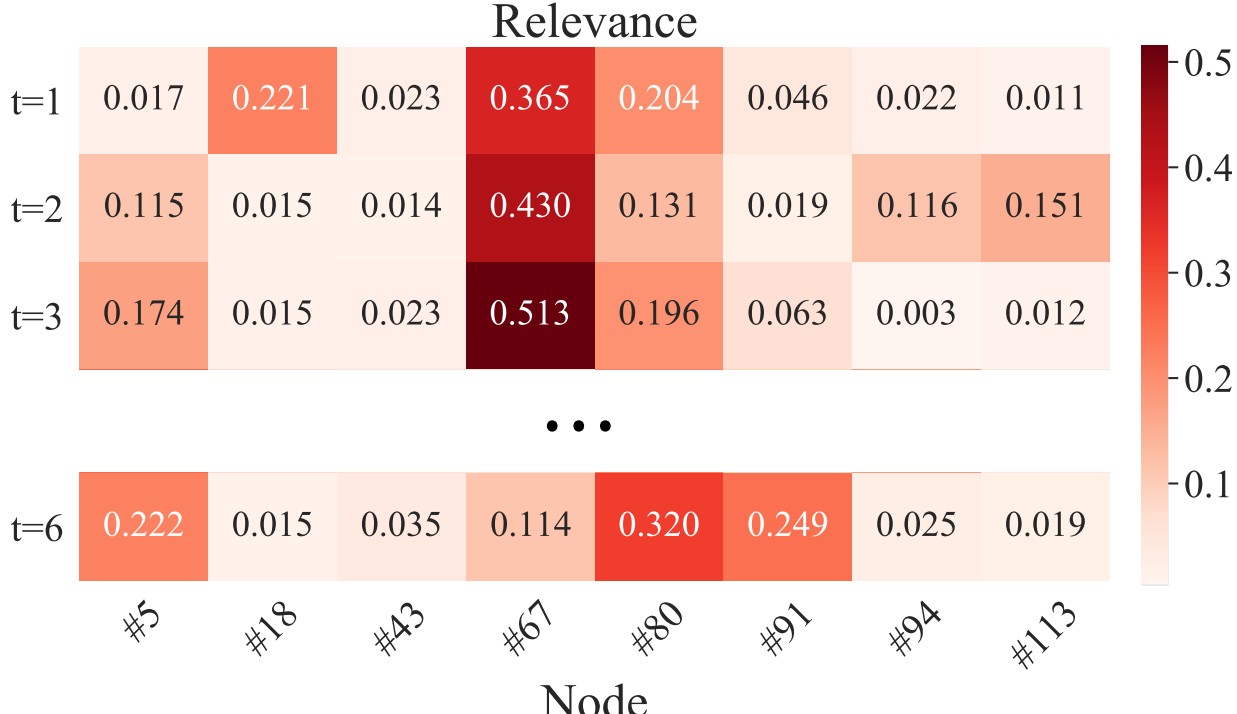

Figure 6: Heatmap of the relevance of eight nodes from time step 1 to 6 on Reddit. Darker colors indicate higher node relevance at each time step.

$\hat{\mathbf{X}}_{t-1}$, without backpropagating the relevance through the GRU layer. As illustrated in Figure 3, Stage 1 is omitted, and the relevance of the hidden representation is directly used as the input for Stage 2. We apply LRP only to the GNN modules at each time step to obtain the relevance of the input features, resulting in $R_{\hat{\mathbf{X}}_t} = R_{\hat{\mathbf{X}}_{t-1}}$. This approach does not account for the relevance back-propagation process within the GRU module, which removes the temporal information needed to explain the dynamic GNNs, specifically in the GCN-GRU model.

Table 4: An ablation study comparing the proposed DGExplainer with the direct application of LRP, which does not account for the temporal information in the GRU module. This experiment evaluates fidelity ($\tau_1 = 0.8$), sparsity ($\tau_2 = 3 \times 10^{-4}$), and stability ($r = 20\%$).

| Method | Metric | Reddit | PeMS04 | PeMS08 | Enron | FB | COLAB |
|---|---|---|---|---|---|---|---|
| DGExplainer | Fidelity ↑ | 0.42 | 0.39 | 0.30 | 0.23 | 0.36 | 0.53 |
| | Sparsity ↑ | 0.87 | 0.99 | 0.95 | 0.85 | 0.96 | 0.96 |
| | Stability ↓ | 0.13 | 0.15 | 0.12 | 0.15 | 0.12 | 0.18 |
| LRP | Fidelity ↑ | 0.29 | 0.27 | 0.25 | 0.20 | 0.30 | 0.48 |
| | Sparsity ↑ | 0.72 | 0.95 | 0.93 | 0.80 | 0.95 | 0.94 |
| | Stability ↓ | 0.18 | 0.21 | 0.16 | 0.12 | 0.17 | 0.19 |

From the results in Table 4, we observe that, generally, LRP exhibits lower fidelity and sparsity compared to DGExplainer. This suggests that DGExplainer, which back-propagates relevance through the GRU, more accurately identifies important features for predictions. Additionally, the stability of DGExplainer is comparable to that of directly applying LRP on the COLAB dataset, and is smaller than many of the baselines in Table 1. This experiment validates the effectiveness of utilizing time-dependent information, which reveals clearer relevance patterns compared to treating each graph independently.

To Revwier 1zv7: Weakness 3

## A.7 Additional Baseline

We conducted additional experiments using GNN-LRP (Schnake et al., 2021), and the results are below:

Table 5: Additional experiments on GNN-LRP evaluated for fidelity ($\tau_1 = 0.8$), fidelity+ ($\tau_1 = 0.8$), sparsity ($\tau_2 = 3 \times 10^{-4}$), and stability ($r = 20\%$).

| Method | Metric | Reddit | PeMS04 | COLAB |
|---|---|---|---|---|
| GNN-LRP | Fidelity ↑ | 0.37 | 0.35 | 0.49 |
| | Fidelity+ ↑ | 0.25 | 0.27 | 0.31 |
| | Sparsity ↑ | 0.82 | 0.99 | 0.74 |
| | Stability ↓ | 0.15 | 0.19 | 0.20 |

From Table 5, we observe that, 1) GNN-LRP exhibits lower fidelity and sparsity compared to DGExplainer. 2) the stability of DGExplainer is higher than that of GNN-LRP, indicating that GNN-LRP is less stable.

To Revwier WFn7: Other points 1

### A.8 Case Study

To demonstrate that the proposed DGExplainer can capture time-varying dependencies among snapshots, we visualize the relevance of several nodes across different time steps using the Reddit dataset, as shown in Figure 6. From this figure, we observe that nodes #43, #67, #80, and #91 maintain relatively consistent importance across time steps 1 to 3, highlighting a continuity in node importance over time within dynamic graphs. Furthermore, node #91 shows lower importance in the earlier snapshots (1 to 3), but its significance increases at time step 6, suggesting that node importance can change over time. To explore the evolution of relevance further, we also visualize the temporal relevance of a specific node, #67, over six consecutive time steps, as shown in Figure 7. We observe that node #67 reaches its peak relevance at time step 3. Its relevance gradually increases before time step 3, begins to decrease at time step 4, and rebounds slightly at time step 5.

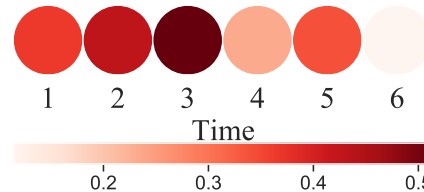

Figure 7: Heatmap of node #67 over six time steps in the Reddit dataset.

To Revwier WFn7: Other points 2

