# OpenReview forum: "DGExplainer: Explaining Dynamic Graph Neural Networks via Relevance Back-propagation"
_TMLR — Rejected by TMLR_

### Review · Reviewer_1zv7 · 2024-08-29

**Summary Of Contributions:**

This paper introduces DGExplainer to explain dynamic GNNs that work on graphs with both temporal and spatial information. DGExplainer is based on layerwise relevance propagation (LRP), where a propagation rule is derived to distribute relevance scores with both temporal and structural features being considered. Extensive experiments have been conducted on multiple dynamic GNNs with a focus on GCN-GRU.

**Audience:**

Yes

**Broader Impact Concerns:**

No.

**Claims And Evidence:**

Yes

**Requested Changes:**

- W1 must be resolved for publication in late 2024.
- Resolving W2 & W3 can strengthen the work.

**Strengths And Weaknesses:**

### Strengths
- The paper is well-motivated and well-written. Explaining dynamic GNNs is an interesting yet under-explored topic.
- The derived rule is straightforward and makes sense.
- Extensive experiments have been done, which clearly demonstrate the effectiveness of DGExplainer over baselines.


### Weaknesses
- The discussed related work and baselines are old (mostly back to 2022).
- Even though the authors have included the performance of DGExplainer for other dynamic GNNs, it would be better to also benchmark the baselines for those GNNs for a more comprehensive understanding.
- Additional ablation studies could be done to further show the importance of capturing temporal info in dynamic graphs.

---

### Review · Reviewer_WFn7 · 2024-09-01

**Summary Of Contributions:**

This work focuses on explaining the behaviour of Dynamic Graph Neural Networks (GNNs), which are a combination of a GNN with a recurrent network (e.g. GRU, or LSTM), with a structure akin to that of a Hidden Markov Model.  Due to the temporal component of the graph, existing explanation methods for static GNNs do not directly extrapolate to this setting. To address this issue, the authors propose DGExplainer, a method that leverages Layer-wise Relevance Propagation (LRP) and apply it to the dynamic GNN setting to propagate the relevance through both time- and space-related features. The authors empirically corroborate the effectiveness of the proposed method, both qualitatively and quantitatively, in a number of datasets, comparing them with previous approaches.

**Audience:**

Yes

**Broader Impact Concerns:**

To some extent, I believe the authors should discuss the usage of explainability methods in real-world scenarios, as well as their weaknesses and potential dangers when applied in the wild. For example, the same author that proposed LRP [showed years later](http://arxiv.org/abs/1906.07983) that the inputs could be manipulated with unperceivable noise so that LRP gave any target output.

**Claims And Evidence:**

No

**Requested Changes:**

The authors should address my three main weaknesses. That is, stating really clearly how does the proposed method differ from just applying LRP on a DGNN (which can be applied automatically to any network using autodiff), and the correctness of the proposed Eq. 3.

**Strengths And Weaknesses:**

**Strengths**
1. The proposed method covers a subfield (dynamic GNNs) for which specific explainability methods are yet underexplored.
2. The paper is well-written and easy to follow.
3. The proposed approach can be of interest to the community, and to practitioners who need explanations over the outputs of their methods.

**Weaknesses**
I have three major concerns with this work, which I would like to bring up:
1. **Correctness**. The authors adopt Eq. 3 as the formula for to compute relevance. While it is said to be inspired by previous works, these other works made sure that the denominator was always non-zero, yet this is not the case in this work. Put differently, as far as I can see there is nothing avoiding the denominator in Eq. 3 to be zero.
2. **Unnecessary derivations**. While I appreciate the hand-made derivations in section 3 (which take 2 pages), I fail to see the advantage of computing them by hand rather than leveraging automatic differentiation tools. What I mean by this is that, assuming that the entries were positive, the rule given in Eq. 3 is simply the chain-rule applied to the logarithm of the output, that is, $\partial \log a_{k2}^{(l+1)}$. After this observation, it is fairly simple to modify the auto-diff operations from, e.g., Pytorch, to automatically compute the all nodes independently of the architecture. Indeed, this was already performed in [this work](https://arxiv.org/abs/2006.03589) that inspired the authors to propose Eq.3.
3. **Contributions**. After the last point, it is quite unclear to me the extent of the contribution of this work, as I am having problems understanding how would this work differ from directly applying LRP to the given DGNN network. Leveraging previous work is totally ok, but I fail to see which difficulties were overcome to leverage LRP in this framework, if any.

Other points that I should highlight:
- Given that the authors drew insights with a paper (the one I linked above) which proposes an LRP-based explainability method, it is a bit surprising that it is missing among the baselines.
- The qualitative analysis is unsatisfactory to my eyes, and not any more clear than the quantitative one. To be clear, for a qualitative analysis I would have liked to see a clearly specific example, where the explanation given is described, and analysed.

---

### Review · Reviewer_wvjk · 2024-09-03

**Summary Of Contributions:**

In this paper, the authors present DGExplainer, a novel and efficient framework that utilizes both layer-wise and time-wise relevance back-propagation to explain the predictions of dynamic Graph Neural Networks (GNNs). To evaluate DGExplainer’s performance, extensive experiments are conducted both quantitative and qualitative experiments. The results demonstrate the framework’s effectiveness in identifying crucial nodes for link prediction and node regression tasks.

**Audience:**

Yes

**Claims And Evidence:**

Yes

**Requested Changes:**

See the weaknesses.

**Strengths And Weaknesses:**

Strengths:
1. This paper proposes a novel framework, DGExplainer, which is designed to generate explanations for dynamic GNNs from a decomposition perspective. DGExplainer effectively calculates relevances that represent the contributions of each component in a dynamic graph.
2. Extensive experiments across three evaluation metrics show that our method provides faithful explanations.

Weaknesses:
1. In the experiments, the authors compare DGExplainer with GNNExplainer, PGExplainer, SubgraphX, TGNNExplainer to evaluate faithfulness. However, the baselines are too old and too weak. Moreover, the authors argue that the proposed method is effective in identifying crucial nodes for link prediction. However, the baselines are explainable GNN methods, which are also not too weak.
2. The metrics are also too weak. To validate the faithfulness, the authors choose Fidelity as the metric. Is it Fidelity+ or Fidelity-? The explanations may be out-of-distribution that may lead the metric itself inaccurate, and there are new metrics in[R1], which can be chosen as a metric to avoid that.
3. In the dynamic GNN models, the authors choose GCN-GRU as the basis with the reason 'recent approaches still do not consistently outperform the GCN-GRU model'. However, there are recent work such as Roland[R2] that outperforms GCN-GRU, which is a better basis.
4. Important related works are missing. There are explainable dynamic GNN works not included [R3].


[R1] Towards Robust Fidelity for Evaluating Explainability of Graph Neural Networks. ICLR 2024

[R2] ROLAND: graph learning framework for dynamic graphs. KDD 2022

[R3] DyExplainer: Explainable Dynamic Graph Neural Networks. arXiv preprint arXiv:2310.16375.

---

### Decision · Action_Editor_QYEs · 2024-10-08

**Recommendation:** Reject

**Comment:**

The paper claims to introduce a novel framework for explainability of dynamic GNNs, though the presented framework can be interpreted as a direct application of LRP (already established for static GNNs), in light of existing computational tools, like autodifferentiation methods. The reviewers are in agreement that the novelty of these results are limited, and I believe the paper will not find a suitable audience among the reads of this journal.

Secondly, the authors were not able to justify why the denominator in equation 18 cannot be 0, and the implications this might have on the proof.

**Audience:**

The framework presented in the paper can be interpreted as directly applying an existing method (LRP) to a slightly different family of neural network architectures. The challenge of this adaptation is not well justified and the closed-form derivations here, in practice, can be approximated with computational techniques such as those used in autodifferentiation. I also believe no deeper insights are provided from the closed-form derivations.

Given that in practice, implementing a version of LRP that would carry over to dynamic GNNs can rely on autodifferentiation techniques, it is my opinion that the results presented here will not find an audience among the readers of this journal.

**Claims And Evidence:**

The paper falls short in terms of supporting evidence for its claims:

- The argumentation that this framework provides novelty over applying LRP to dynamic GNNs, in light of existing computational tools for back-propagation, is not supported by sufficient evidence.

- Furthermore, it is unclear how the math holds up given that it is not obvious why the denominator in equation 13 (the relevance back-propagation rule of the algorithm) cannot be 0.